# Linkage of Maternity Hospital Episode Statistics data to birth registration and notification records for births in England 2005-2014: methods. A population-based birth cohort study

Nirupa Dattani,[1] Alison Macfarlane[2]

## ABSTRACT

**Introduction** Maternity Hospital Episode Statistics (HES) data for 2005–2014 were linked to birth registration and birth notification data (previously known as NHS Numbers for Babies or NN4B) to bring together some key demographic and clinical data items not otherwise available at a national level. The linkage algorithm that was previously used to link 2005–2007 data was revised to improve the linkage rate and reduce the number of duplicate HES records.

**Methods** Birth registration and notification linked records from the Office for National Statistics ('ONS birth records') were further linked to Maternity HES delivery and birth records using the NHS Number and other direct identifiers if the NHS Number was missing.

**Results** For the period 2005–2014, over 94% of birth registration and notification records were correctly linked to HES delivery records. Two per cent of the ONS birth records were incorrectly linked to the HES delivery record and 5% of ONS birth records were linked to more than one HES delivery record. Therefore, a considerable amount of time was spent in quality assuring these files.

**Conclusion** The linkage rate for birth registration and notification records to HES delivery records steadily improved from 2005 to 2014 due to improvement in the quality and completeness of patient identifiers in both HES and birth notification data.

## Strengths and limitations of this study

► Linking three national data on births together greatly increased the number of variables available for analysis.
► The findings are relevant for other users of trusted third party linkage who should not assume that datasets linked using patient identifiers are error free and may affect any analyses carried out on them.
► Data are held in a secure environment at the Office for National Statistics so access is restricted but can be used by approved researchers.

## INTRODUCTION

When a baby is born in England and Wales, data are recorded in several separate information systems, namely birth registrations where mainly socio-demographic data are collected. A smaller set of data is recorded when the birth is notified to the NHS and the NHS Number, a unique identifier, is issued. Data about care at delivery are recorded in the Hospital Episode Statistics (HES) if the birth occurs in England. Data about care at delivery in Wales are recorded in the Patient Episode Database for Wales which is linked to the National Community Child Health database. Each of these systems includes common data items such as the baby's and mother's date of birth, postcode of residence and NHS Number which can be used as identifiers for record linkage.

In England and Wales, all live births must be registered within 42 days. The data recorded at registration include names, address of residence, place of birth, occupation of the parents and country of birth of mother and the father.[1] The introduction of the interim NHS Numbers for Babies (NN4B) Service in 2002 provided the opportunity to obtain information such as gestational age and baby's ethnicity data. Information on gestational age at birth is of key importance as babies born preterm, before 37 completed weeks of gestation, are at particularly high risk of morbidity and mortality in early years of life.[2–4]

A collaborative project was set up in 2004 between City University London, the Office for National Statistics (ONS) and the Welsh Assembly to link these datasets for all births that occurred in England and Wales from 2005 to 2007. Stage 1 of the



[1]Centre for Maternal and Child Health Research, City, University of London, London, UK
[2]Centre for Maternal and Child Health Research, City, University of London, London, UK

**Correspondence to**
Ms Nirupa Dattani;
n.dattani.1@city.ac.uk

**Table 1** Availability of selected data items from birth registration, birth notification and maternity HES

| Data items | Data sources | | |
| --- | --- | --- | --- |
| | Birth registration | Birth notification | Maternity HES |
| Baby's NHS Number | + | + | + |
| Mother's NHS Number | − | + | + |
| Birth date of baby | + | + | + |
| Delivery time | − | + | − |
| Birth weight | + | + | + |
| Gestational age (stillbirth) | + | + | + |
| Gestational age (live birth) | − | + | + |
| Sex of baby | + | + | + |
| Number of babies born | + | + | + |
| Live or stillbirth | + | + | + |
| Parity (all births) | − | − | + |
| Baby/mother's postcode of usual residence | + | + | + |
| Ethnic category of baby | − | + | − |
| Ethnic category of mother | − | − | + |
| Country of birth of mother | + | − | − |
| Country of birth of father | + | − | − |
| Father's socioeconomic status | + | − | − |
| Type of delivery place | + | + | + |
| Mother's date of birth | + | + | + |
| Marital status of mother | + | − | − |
| Method of delivery | − | − | + |
| Complications in pregnancy | − | − | + |

HES, Hospital Episode Statistics.

project involved linkage of birth registration data with the birth notification data (previously known as NN4B dataset) and assessment of the quality and completeness of the notification data. This was piloted on the 2005 data.[5] [6] Since 2007, these datasets have been routinely linked by ONS and gestation-specific infant mortality and birth statistics have been published annually.[7]

Stage 2 of the project involved linkage of the dataset created in stage 1 to Maternity HES and assessment of data quality and completeness by comparison with birth registration or notification dataset, where possible.[8] The linkage of Welsh data for all three years (2005–2007) was carried out separately.[9]

The primary focus of the first two projects was to test the feasibility of the linkages and assess the quality of the linked datasets. The next project aimed to answer a specific set of research questions. In 2013, a project was funded to describe and analyse daily, weekly and yearly cyclical variations in births and their outcome and explore the potential implications of the patterns observed for NHS staffing and for service users. This involved extension of the linkage in stages 1 and 2 to include births occurring in England and Wales between 1 January 2005 and 31 December 2014. This article describes the linkage of data for England. As before, data for Wales were linked separately.

Several variables are common to all three data sources, Maternity HES, birth registration and birth notification, as can be seen in table 1. In addition, some data items are unique to each data source and linkage enables new analyses using these linked data. For example, it is now possible to compare time of birth with birth outcomes, and report on the outcomes of birth by care at birth in terms of onset of labour and mode of delivery by gestational age, time of day and day of the week.

## METHODS
### Data sources

The source data, birth registration, birth notification and Maternity HES are described in detail in the article describing the linkage of data for 2005 and 2006.[8]

There are two types of maternity records in HES: the delivery record and the birth record. Both types of records consist of an admitted patient care record with an additional 19 fields, in an appended baby 'tail'.

The HES delivery record is a mother-based record containing the mother's details with a maternity tail and a baby tail which can accommodate up to nine babies born in one maternity. In contrast, the birth registration and notification linked data consist of one record per baby. Therefore, the linkage was based on linking babies to their mothers' records.

A HES birth record is generated for the baby. It contains the baby's details and also has a baby tail containing the same type of information that is recorded in the corresponding baby tail of the mother's delivery record.

The baby tail data coverage is less complete than the rest of the HES data. There are a number of reasons for the incompleteness and data quality issues, such as
► trusts submitting a significantly higher number of delivery episodes compared with birth episodes
► trusts failing to submit data on the number of birth episodes where they record a higher number of delivery episodes.

**Table 2** Number and percentage of birth registration and notification linked records linked to HES delivery records, England, 2005–2014

| Year of birth | Number of ONS birth records | Number of ONS birth records linked to HES delivery records (excluding duplicate HES delivery records) | Number of duplicate HES delivery records (ie, more than one HES delivery record per ONS birth record) | Never linked to HES delivery record | Linkage rate | Linked HES delivery records after quality assurance process | Percentage linked after quality assurance |
|---|---|---|---|---|---|---|---|
| 2005 | 617 613 | 582 963 | 25 188 | 34 650 | 94.4 | 571 775 | 92.6 |
| 2006 | 640 271 | 607 649 | 23 582 | 32 622 | 94.9 | 592 028 | 92.5 |
| 2007 | 659 061 | 632 039 | 27 207 | 27 022 | 95.9 | 614 542 | 93.2 |
| 2008 | 676 999 | 655 511 | 24 192 | 21 488 | 96.8 | 640 900 | 94.7 |
| 2009 | 675 330 | 657 622 | 40 575 | 17 708 | 97.4 | 642 508 | 95.1 |
| 2010 | 687 100 | 673 566 | 50 086 | 13 534 | 98.0 | 662 014 | 96.3 |
| 2011 | 688 681 | 674 751 | 45 005 | 13 930 | 98.0 | 663 135 | 96.3 |
| 2012 | 698 457 | 681 677 | 41 373 | 16 780 | 97.6 | 668 055 | 95.6 |
| 2013 | 668 433 | 651 957 | 42 656 | 16 476 | 97.5 | 641 108 | 95.9 |
| 2014 | 664 967 | 647 047 | 46 932 | 17 920 | 97.3 | 635 692 | 95.6 |
| Total | 6676 912 | 6464 782 | 366 796 | 212 130 | 96.8 | 6331 757 | 94.8 |

HES, Hospital Episode Statistics; ONS, Office for National Statistics.

## Record linkage

Patient identifiers including mother's and baby's NHS numbers, postcode of residence, mother's and baby's date of birth and baby's sex together with a unique record ID were extracted by ONS from the linked birth registration and notification file and sent to the data linkage team at the Health and Social Care Information Centre, now known as NHS Digital.

The linkage algorithm that had been previously used to link 2005–2007 births was used to link further years data 2008–2014. ONS identifiers were first linked to the HES index to obtain HES patient identifiers known as HESIDs.[10] These were then linked to the HES delivery records, but the number of duplicate HES delivery records linked to ONS birth records was very much higher than it had been when the data for 2005–2007 were linked. NHS Digital therefore recommended using its inhouse linkage algorithm that is used routinely to link ONS death registration data to HES[11] except in our study step 8 of the algorithm involved using only the NHS Number, as shown in online supplementary appendix A. This was piloted on the 2005 data and the number of duplicate HES records linked to the ONS birth record and the linkage rate was ascertained before data for 2006–2014 were linked.

The linked data provided by NHS Digital consisted of two files for each financial year from 2004/2005 to 2014/2015. One file contained ONS birth records linked to the HES delivery records and a second file, based on linkage of ONS birth records to HES baby records.

The linked data were accessed by researchers from City, University of London in the secure setting of the Virtual Microdata Laboratory facility at ONS. The researchers concerned had ONS Approved Researcher status.

The quality of linkage was assessed to ensure that the ONS birth record was linked to the correct delivery record in HES. This involved use of deterministic stepwise rules based on a combination of data items common to both datasets such as place of birth, birthweight, date of birth of the baby, gestational age, multiplicity and sex of baby.

## RESULTS
### Mother file

A pilot study was carried out using the 2005 data. The file sent to NHS Digital consisted of 617 613 babies who were either born in England or resident in England. The resident in England category was used for births that occurred at home in the ONS linked dataset. NHS Digital first linked these to the HES index to get the HESID and then to the HES delivery records. The file returned to ONS consisted of 624 326 records with a HESID and a second file of 582 963 of ONS birth records that were linked to the HES delivery record. The number of ONS births linked to HESID was higher as it included old and new HESID for some women. This normally happens when a woman is allocated a new HESID and it subsequently becomes evident that she has already been assigned a HESID previously. In addition, there were 25 188 duplicate HES delivery records, that is where the ONS birth record was linked to more than one HES delivery record (table 2). By using the revised linkage algorithm, the number of duplicate HES delivery records linked to ONS birth records was reduced to 4% from 6%.[8] Data for 2006–2014 were

**Table 3** Percentages of ONS birth records linked to HES delivery records by match rank, England, 2005–2014

| Year of birth | Match rank | | | | | | | | Total |
|---|---|---|---|---|---|---|---|---|---|
| | 1 | 2 | 3 | 4 | 5 | 6 | 7 | 8 | |
| 2005 | 66.0 | 2.6 | 1.0 | 0.1 | 0.9 | 28.8 | 0.6 | 0.1 | 100.0 |
| 2006 | 69.4 | 2.6 | 1.0 | 0.1 | 1.2 | 25.1 | 0.5 | 0.1 | 100.0 |
| 2007 | 73.5 | 2.9 | 1.2 | 0.1 | 0.5 | 21.3 | 0.4 | 0.0 | 100.0 |
| 2008 | 77.3 | 2.9 | 1.3 | 0.1 | 0.4 | 17.6 | 0.4 | 0.0 | 100.0 |
| 2009 | 81.7 | 2.8 | 1.4 | 0.1 | 0.3 | 13.4 | 0.3 | 0.0 | 100.0 |
| 2010 | 85.5 | 2.5 | 1.5 | 0.1 | 0.3 | 10.0 | 0.2 | 0.0 | 100.0 |
| 2011 | 88.2 | 2.5 | 1.5 | 0.1 | 0.3 | 7.2 | 0.2 | 0.0 | 100.0 |
| 2012 | 90.5 | 2.5 | 1.6 | 0.1 | 0.3 | 4.9 | 0.1 | 0.0 | 100.0 |
| 2013 | 92.0 | 2.5 | 1.7 | 0.1 | 0.2 | 3.5 | 0.1 | 0.0 | 100.0 |
| 2014 | 92.7 | 2.7 | 1.8 | 0.1 | 0.3 | 2.5 | 0.1 | 0.0 | 100.0 |

HES, Hospital Episode Statistics; ONS, Office for National Statistics.

therefore then linked to HES delivery records using the revised linkage algorithm shown in online supplementary appendix A.

Around 66% of the previously linked ONS birth registration and notification records were linked to the HES delivery records in stage 1 of the linkage algorithm shown in table 3. This matched records having same mother's NHS Number, exact date of birth, sex and exact full postcode. A further 29% of the ONS birth records were matched to HES delivery records using the exact date of birth, postcode of residence of the mother and sex (stage 6 of the algorithm). About 5% of the records were linked using a combination of mother's NHS Number, exact or partial date of birth, sex and

**Table 4** Number of ONS birth records linked to HES birth records, England, 2005–2014

| Year of birth | Number of ONS Birth records | Number of ONS birth records linked to HES birth records | Percentage linked |
|---|---|---|---|
| 2005 | 617 613 | 609 778 | 98.73 |
| 2006 | 640 271 | 633 183 | 98.89 |
| 2007 | 659 061 | 651 551 | 98.86 |
| 2008 | 676 999 | 668 967 | 98.81 |
| 2009 | 675 330 | 669 926 | 99.20 |
| 2010 | 687 100 | 682 261 | 99.30 |
| 2011 | 688 681 | 683 768 | 99.29 |
| 2012 | 698 457 | 693 221 | 99.25 |
| 2013 | 668 433 | 662 963 | 99.18 |
| 2014 | 664 967 | 659 192 | 99.13 |

HES, Hospital Episode Statistics; ONS, Office for National Statistics.

postcode. ONS birth records that were not linked to HES accounted for 3% of all records.

Linkage of ONS birth records to HES delivery records for births from 2005 to 2014 showed that the number of records linked using stage 1 of the algorithm increased from 66% in 2005 to 93% in 2014. There was a corresponding decrease in the number of records linked in stage 6 of the algorithm which excludes use of mother's NHS Number from 29% to 3%.

Each year there were about 36 000 duplicate HES delivery records linked to ONS birth records, that is, where the same ONS birth record was linked to multiple HES delivery records. During assessment of the quality of linkage, the HES delivery record with mother's and baby's information matching the ONS birth record and with greatest amount of information on onset of labour and method of delivery was retained for analysis. The other records were discarded. In addition, there were 13 300 HES delivery records incorrectly linked to the ONS birth records. It took over 71 days to assess the quality of linkage and to produce a final linked dataset consisting of one ONS birth record linked to the relevant HES delivery record.

### Baby file
The baby file was much more straightforward to link than the mother file as it involved a one-to-one link between an ONS birth record and a HES birth record, also referred to as the HES baby record.

The numbers of HES birth records linked to ONS birth records, for each year from 2005 to 2014, were higher than the numbers of HES delivery records linked to the ONS birth records (see table 4). The quality of linkage of the baby file has yet to be assessed.

### Linkage bias
Although the linkage rate increased from 94% in 2005 to 97% in 2014, there were statistically significant

**Table 5A** All births in England linked to delivery HES records by year of birth, 2005–2014.

| Year of birth | Number of ONS birth records | Linked to HES delivery record | Never linked to HES delivery record | Linkage rate |
|---|---|---|---|---|
| 2005 | 617 613 | 582 963 | 34 650 | 94.39 |
| 2006 | 640 271 | 607 649 | 32 622 | 94.90 |
| 2007 | 659 061 | 632 039 | 27 022 | 95.90 |
| 2008 | 676 999 | 655 511 | 21 488 | 96.83 |
| 2009 | 675 330 | 657 622 | 17 708 | 97.38 |
| 2010 | 687 100 | 673 566 | 13 534 | 98.03 |
| 2011 | 688 681 | 674 751 | 13 930 | 97.98 |
| 2012 | 698 457 | 681 677 | 16 780 | 97.60 |
| 2013 | 668 433 | 651 957 | 16 476 | 97.54 |
| 2014 | 664 967 | 647 047 | 17 920 | 97.31 |
| Total | 6 676 912 | 6 464 782 | 212 130 | 96.82 |

39689.65073 Pearson $\chi^2$ statistic.
9 <- df = (rows−1)×(colums− 1).
0.0000 <- getting the P value from the $\chi^2$ statistics and the df.
HES, Hospital Episode Statistics; ONS, Office for National Statistics.

**Table 5B** All births in England linked to delivery HES records by multiplicity, 2005–2014.

| Sex of baby | Number of ONS birth records | Linked to HES delivery record | Never linked to HES delivery record | Linkage rate |
|---|---|---|---|---|
| Singletons | 6 468 586 | 6 268 013 | 200 573 | 96.90 |
| Multiples | 208 326 | 196 769 | 11 557 | 94.45 |
| Total | 6 676 912 | 6 464 782 | 212 130 | 96.82 |

3928.081999 Pearson $\chi^2$ statistic.
1 <- df = (rows−1)× (columns−1).
0.0000 <- getting the P value from the $\chi^2$ statistics and the df.
HES, Hospital Episode Statistics; ONS, Office for National Statistics.

**Table 5C** All births in England linked to delivery HES records by age of mother, 2005–2014.

| Age of mother | Number of ONS birth records | Linked to HES delivery record | Never linked to HES delivery record | Linkage rate |
|---|---|---|---|---|
| Under 15 | 1739 | 1653 | 86 | 95.05 |
| 15–19 | 206 936 | 200 367 | 6569 | 96.83 |
| 20–24 | 1 218 562 | 1 186 178 | 32 384 | 97.34 |
| 25–29 | 1 812 830 | 1 764 402 | 48 428 | 97.33 |
| 30–34 | 1 926 290 | 1 865 427 | 60 863 | 96.84 |
| 35–39 | 1 092 622 | 1 048 332 | 44 290 | 95.95 |
| 40–44 | 245 526 | 232 397 | 13 129 | 94.65 |
| 45 and more | 15 821 | 14 020 | 1801 | 88.62 |
| Not stated | 156 586 | 152 006 | 4580 | 97.08 |
| Total | 6 676 912 | 6 464 782 | 212 130 | 96.82 |

12579.68484 Pearson $\chi^2$ statistic.
8 <- df = (rows−1)× (columns− 1).
0.0000 <- getting the P value from the $\chi^2$ statistics and the df.
HES, Hospital Episode Statistics; ONS, Office for National Statistics.

**Table 5D** All births in England linked to delivery HES records by ethnicity, 2005–2014.

| Ethnicity of baby | Number of ONS birth records | Linked to HES delivery record | Never linked to HES delivery record | Linkage rate |
|---|---|---|---|---|
| Bangladeshi | 93 074 | 91 081 | 1993 | 97.86 |
| Indian | 199 963 | 194 212 | 5751 | 97.12 |
| Pakistani | 272 457 | 266 007 | 6450 | 97.63 |
| Black African | 215 621 | 203 962 | 11 659 | 94.59 |
| Black Caribbean | 65 048 | 62 685 | 2363 | 96.37 |
| White British | 4 239 203 | 4 140 349 | 98 854 | 97.67 |
| White other | 547 384 | 523 826 | 23 558 | 95.70 |
| Other | 628 556 | 602 162 | 26 394 | 95.80 |
| Not stated | 415 606 | 380 498 | 35 108 | 91.55 |
| Total | 6 676 912 | 6 464 782 | 212 130 | |

61433.75188 Pearson $\chi^2$ statistic.
8 <- df = (rows−1)× (colums−1).
0.0000 <- getting the P value from the $\chi^2$ statistics and the df.
Not stated includes ethnicity ticked as 'not known' and missing.
HES, Hospital Episode Statistics; ONS, Office for National Statistics.

**Table 5E** All births in England linked to delivery HES records by sex, 2005–2014.

| Baby's sex | Number of ONS birth records | Linked to HES delivery record | Never linked to HES delivery record | Linkage rate |
|---|---|---|---|---|
| Female | 3 253 584 | 3 149 797 | 103 787 | 96.81 |
| Male | 3 423 327 | 3 314 985 | 108 342 | 96.84 |
| Not stated | 1 | 0 | 1 | 0.00 |
| Total | 6 676 912 | 6 464 782 | 212 130 | 96.82 |

33.89542247 Pearson $\chi^2$ statistic.
2 <- df = (rows−1)× (columns−1).
0.0000 <- getting the P value from the $\chi^2$ statistics and the df.
HES, Hospital Episode Statistics; ONS, Office for National Statistics.

**Table 5F** All births in England linked to delivery HES records by gestational age, 2005–2014.

| Gestational age | Number of ONS birth records | Linked to HES delivery record | Never linked to HES delivery record | Linkage rate |
|---|---|---|---|---|
| Missing or less than 22 weeks | 53 236 | 50 420 | 2816 | 94.71 |
| Preterm | 506 206 | 486 517 | 19 689 | 96.11 |
| Term | 5 861 275 | 5 678 501 | 182 774 | 96.88 |
| Post-term | 256 195 | 249 344 | 6851 | 97.33 |
| Total | 6 676 912 | 6 464 782 | 212 130 | 96.82 |

1884.065581 Pearson $\chi^2$ statistic.
3 <- df = (rows− 1)× (columns− 1).
0.0000 <- getting the P value from the $\chi^2$ statistics and the df.
HES, Hospital Episode Statistics; ONS, Office for National Statistics.

differences between distributions of records that were linked to HES delivery records by NHS Digital and those that were not linked in terms of multiplicity, age of mother, ethnicity and region of residence (table 5). The linkage rate was 3% lower for multiple births than for singletons, 2% lower in mother's aged under 15 years and 2% lower for those aged 40 years and above compared with all other age groups. A comparison by baby's ethnicity showed that over 5% of black African

**Table 5G** All births in England linked to delivery HES records by time of delivery, 2005–2014.

| Hour of birth | Number of ONS birth records | Linked to HES delivery record | Never linked to HES delivery record | linkage rate |
|---|---|---|---|---|
| 0.00–2.59 | 816647 | 791373 | 25274 | 96.91 |
| 3.00–5.59 | 801801 | 776911 | 24890 | 96.90 |
| 6.00–8.59 | 711622 | 687834 | 23788 | 96.66 |
| 9.00–11.59 | 1094422 | 1061279 | 33143 | 96.97 |
| 12.00–14.59 | 869441 | 840948 | 28493 | 96.72 |
| 15.00–17.59 | 795151 | 769329 | 25822 | 96.75 |
| 18.00–20.59 | 743832 | 719896 | 23936 | 96.78 |
| 21.00–23.59 | 782848 | 758454 | 24394 | 96.88 |
| Not stated | 61148 | 58758 | 2390 | 96.09 |
| Total | 6676912 | 6464782 | 212130 | 96.82 |

334.9624025 Pearson $\chi^2$ statistic.
8 <- df = (rows–1)× (columns– 1).
0.0000 <- getting the P value from the $\chi^2$ statistics and the df.
HES, Hospital Episode Statistics; ONS, Office for National Statistics.

**Table 5H** All births in England linked to delivery HES records by region of usual residence, 2005–2014.

| Region | Number of ONS birth records | Linked to HES delivery record | Never linked to HES delivery record | Linkage rate |
|---|---|---|---|---|
| East Midlands | 456324 | 448489 | 7835 | 98.28 |
| East of England | 672006 | 652543 | 19463 | 97.10 |
| London | 1298130 | 1227661 | 70469 | 94.57 |
| North East | 307532 | 301141 | 6391 | 97.92 |
| North West | 867881 | 850726 | 17155 | 98.02 |
| South Central | 462848 | 455243 | 7605 | 98.36 |
| South East Coast | 514289 | 503042 | 11247 | 97.81 |
| South West | 566860 | 559043 | 7817 | 98.62 |
| West Midlands | 709445 | 695469 | 13976 | 98.03 |
| Yorkshire/The Humber | 645649 | 635293 | 10356 | 98.40 |
| Elsewhere | 10989 | 9507 | 1482 | 86.51 |
| Home | 164954 | 126622 | 38332 | 76.76 |
| Not stated | 5 | 3 | 2 | 60.00 |
| Total | 6676912 | 6464782 | 212130 | 96.82 |

269313.4278 Pearson $\chi^2$ statistic.
12 <df = (rows–1)× (columns– 1).
0.0000 <- getting the P value from the $\chi^2$ statistics and the df.
HES, Hospital Episode Statistics; ONS, Office for National Statistics.

babies were not linked to Maternity HES. Over 98% of the babies resident in East Midlands, North West, South Central, South West, West Midlands, and Yorkshire and The Humber were successfully linked to Maternity HES, and this proportion was slightly lower, 95%, among babies resident in London.

## DISCUSSION
Although the data linkage team at NHS Digital has experience of linking external datasets to HES and we used a similar linkage algorithm to that routinely used by NHS Digital to link ONS death records to HES records, there were issues with the quality of linkage. In the period 2005–2014, 2% of HES delivery records were incorrectly linked to the ONS birth records as common data items such as place of birth, date of birth of baby, gestational age, birthweight, multiplicity and sex differed in the HES delivery and ONS birth records. In addition, 366000 duplicate HES delivery records were linked to ONS birth records. This meant that a considerable amount of time was spent in quality assuring these files.

The number of birth registration and notification records linked to the HES delivery records using the NHS Number increased over the years from 2005 to 2014. This was not surprising as completeness of the mother's NHS Number improved over time in the registration and notification linked records. In 2005, the mother's NHS Number was present in over two-thirds of the records and this increased to over 90% in 2014. There were also a small proportion of HES records that had the mother's NHS Number missing. A further quarter of the registration and notification linked records in 2005 were linked using exact date of birth, sex and postcode which reduced to 3% in 2014. There were concerns about using postcode in the linkage algorithm for linking data for earlier years, as the HES index may not hold all historical postcodes of residence of the mother and the postcode on registration and notification linked data was recorded at the time of registration. It is possible the mother could have moved since having the baby and this variable is also subject to recording and reporting errors.

Overall, a linkage rate of over 90% was achieved and it improved over time, especially in 2014, when there had been a shorter time before linkage was carried out and HESID would have been less likely to have changed. This suggests that HESID at birth could be retained as a separate field for linkage.

Although the linkage rate for ONS birth records to HES births was higher than the linkage rate for the delivery records and we did not assess the quality of linkage, our previous linkage study showed that there were many duplicate HES birth records linked to ONS birth records.[8] In addition,

NHS Digital acknowledges that a high proportion of baby records are known to be missing in Maternity HES.[12] HES delivery records include information about the baby and the mother so the quality of information in HES was assessed using the delivery records.

While ONS birth registration data have remained of consistently high quality, there have been issues with data quality and completeness in Maternity HES.[8,12,13] The number of births and deliveries in London are under-represented in Maternity HES which could be due to under-reporting or complete lack of reporting, of births by several hospitals. Also HES currently captures few home births and none occurring in private hospitals, even though data about all births should be submitted to Maternity HES.

## CONCLUSIONS

This study shows that it is possible to link a large majority of the linked birth registration and notification records to Maternity HES records, but linkage would be considerably more valuable if data quality and completeness improved in Maternity HES. Information about parity, onset of labour, method of delivery and complications in pregnancy can only be obtained at a national level from Maternity HES, so linking all three national datasets on births and maternity would expand the scope and range of data available.

**Acknowledgements** The authors would like to thank all the relevant colleagues in the Office for National Statistics, NHS Digital, formerly the Health and Social Care Information Centre and the NHS Wales Informatics Service for their, help. In particular, they would like to thank Emma Gordon, Joanne Evans, Claudia Wells, Alex Lloyd, Justine Pooley, Elizabeth Mclaren and members of the VML Team at the Office for National Statistics, Ariane Alamdari and Garry Coleman at NHS Digital and Gareth John at the NHS Informatics Service. The authors would also like to thank everyone who took part in our Public and Patient Involvement activities for the advice and insights they gave and the members of our Study Advisory Group for their help and advice.

**Contributors** Nirupa Dattani was responsible for the linkage methodology and writing first draft of the paper, and Professor Alison Macfarlane re-drafted and provided final approval of the version to be published. We agree to be accountable for all aspects of the work in ensuring that questions related to the accuracy or integrity of any part of the work are appropriately investigated and resolved.

**Funding** This work was supported by the National Institute for Health Research. HS&DR Programme, project number HS&DR 12/136/93. Project title: 'Births and their outcome: analysing the daily, weekly and yearly cycle and their implications for the NHS'.

**Disclaimer** The views and opinions expressed here are those of the authors and do not necessarily reflect those of the HS&DR programme, NIHR, NHS or the Department of Health. The data were processed in the secure environment of the Office for National Statistics' Virtual Microdata Laboratory and the following disclaimer applies: This work contains statistical data from ONS which is Crown Copyright. The use of the ONS statistical data in this work does not imply the endorsement of the ONS in relation to the interpretation or analysis of the statistical data. This work uses research datasets which may not exactly reproduce National Statistics aggregates.

**Competing interests** None declared.

**Patient consent** Not required.

**Ethics approval** East London and City Local Research Ethics Committee 1 and its successors.

**Provenance and peer review** Not commissioned; externally peer reviewed.

**Data sharing statement** The authors do not have permission to supply data or identifiable information to third parties, including other researchers but they have permission under Section 251 of the Health Service (Control of Patient Information) Regulations 2002 to analyse patient identifiable data for England and Wales without consent and create a research database which could be accessed by other researchers using the VML at the Office for National Statistics. Anyone wishing to access the linked datasets for research purposes should apply to the Office for National Statistics and NHS Digital as well as to the Confidentiality Advisory Group of the Health Research Authority to access patient identifiable data without consent. We are currently in discussion with ONS and NHS Digital about the application process.

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
