## [Reviewer comments · BMJ Open]

ARTICLE DETAILS

TITLE (PROVISIONAL)	Linkage of Maternity Hospital Episode Statistics data to birth registration and NHS Numbers for babies records in England 2005-2014: methods. A population based birth cohort study.
AUTHORS	Dattani, Nirupa; Macfarlane, Alison

VERSION 1 – REVIEW

REVIEWER	Rachael Wood NHS National Services Scotland Information Services Division Scotland UK I am responsible for maintaining the linkage of maternal and baby health records in Scotland but have had no involvement in this English work.
REVIEW RETURNED	10-Jun-2017

GENERAL COMMENTS	This paper reports on the updated linkage of birth statutory registration and NHS notification records to HES delivery and birth records. All births in England 2005-2014 are included. This is important work of interest and benefit to the maternity and child health research community generally. In general the paper is clear and well presented. Addressing the following comments would bring further benefit. Abstract The second paragraph of the results section introduces findings that are not in the main paper. If the authors wish to present these findings, they should add a substantive table showing completeness of key data items in different data sources. Introduction This is clear. Methods The second paragraph of the record linkage section suggests that step 8 of the usual NHS Digital linkage algorithm was omitted for this project but this does not tally with Table 2: please clarify. Can a reference be provided for the paper on quality assuring the linked dataset?
---

	Results In the first paragraph of the mother file section can the authors clarify why NHS Digital returned a higher number of births linked to HESID (624,326) than there were in the birth registration/notification file (617,613)? In the second paragraph of the baby file section, the authors state that a lower proportion of birth registration/notification records linked to HES birth records than to HES delivery records but this does not tally with Tables 1 and 3. In the linkage bias section, the authors state that >2% of records relating to Black African babies did not link to HES delivery records but Table 4 suggests it was >5%. Discussion In the first paragraph the authors state that 4-6% of birth registration/notification records were 'incorrectly linked' to HES delivery records. Can this statement be clarified? A reference 14 is cited but this is missing from the reference list. Conclusions The second paragraph may well be true but it is not supported by results presented in this paper. Please either provide supporting results or amend. Tables Should the heading of the 3rd column in Table 1 read including (rather than excluding) duplicated HES delivery records? In general the headings of the tables could be tidied up and made more consistent. For example, Table 3 does not provide results by match rank and the terminology for specific datasets is variable eg birth notification/NN4B. No reporting checklist accompanies this paper as far as I can tell. A STROBE or, preferably, a RECORD statement would be appropriate.
--	--

REVIEWER	Toan C Ong University of Colorado Anschutz Medical Campus, Aurora, Colorado USA No Competing Interest
REVIEW RETURNED	14-Jun-2017

GENERAL COMMENTS	Thank you for the opportunity to review this paper. This paper describes the efforts to perform linkage between clinical and demographics data to ultimately generate a more complete dataset of mothers and babies. It is very encouraging to learn that the authors were able to link large datasets using a small set of overlapping linkage variables. While there are very interesting insights from the results, there are major concerns: - The value of linking real patient data is apparent. However, that alone will be insufficient to justify a separate publication. I struggled to see the scientific contributions of this paper.
---

	This entire paper can fit into the Result and Discussion of a more comprehensive paper.  - Other than the description of the linkage variables and how they were compared (exact or partial), the record linkage method was not adequately described. - It was not clear how the linkage result was validated. The authors mentioned that process is described elsewhere but I disagree with that separation. - Many observations were described vaguely without further articulation. For example, quote from the paper "The linkage rate was 3 per cent lower for multiple births, 2 per cent lower in mother's aged under 15 and 3 per cent lower for those aged 40 years and above." Questions: Lower than what? Why lower? - Using the NHS number helped link the majority of the datasets. It is interesting to know how many records were linked using just the NHS number as the linkage variable. What is the performance when the NHS is absent? Overall, I find that the findings of this paper are very interesting. However, the authors should consider a comprehensive paper which discusses in details the linkage methods, the results, and the validation process. Such paper will be much more impactful than separate ones.
--	---

VERSION 1 – AUTHOR RESPONSE

Reviewer Name: Rachael Wood

Abstract

Comment: The second paragraph of the results section introduces findings that are not in the main paper. If the authors wish to present these findings, they should add a substantive table showing completeness of key data items in different data sources.

Response: Thanks for pointing this out. These findings were identified with the 2005 to 2007 that we linked previously and the paper was published in the ONS Journal Health Statistics Quarterly. I have now removed these findings from the abstract.

Methods

Comment: The second paragraph of the record linkage section suggests that step 8 of the usual NHS Digital linkage algorithm was omitted for this project but this does not tally with Table 2: please clarify.

Response: Sorry to have confused you. I have now re-written this paragraph to say that step 8 of the linkage algorithm was used in our study.

Comment: Can a reference be provided for the paper on quality assuring the linked dataset?

Response: We were hoping that the paper by Gill Harper titled 'Quality assuring linked ONS birth records and maternity Hospital Episode Statistics delivery records on singleton and multiple births in England 2005 to 2014' would be published by BMJ Open at the same time as this linkage paper but this may not happen so I have mentioned briefly in this paper how the linkage was quality assured and how long it took.

Results

Comment: In the first paragraph of the mother file section can the authors clarify why NHS Digital returned a higher number of births linked to HESID (624,326) than there were in the birth registration/notification file (617,613)?

Response: There were a higher number of births linked to HESID as it included old and new HESID for some women. This normally happens when a woman is allocated a new HESID and it subsequently becomes evident that the woman has already been assigned a HESID previously.

Comment: In the second paragraph of the baby file section, the authors state that a lower proportion of birth registration/notification records linked to HES birth records than to HES delivery records but this does not tally with Tables 1 and 3.

Response: Thanks for picking this up. You are correct that the proportion of birth registration/notification records linked to HES birth records is higher than HES delivery records so the text has now been corrected.

Comment: In the linkage bias section, the authors state that >2% of records relating to Black African babies did not link to HES delivery records but Table 4 suggests it was >5%.

Response: Thanks for identifying this. You are absolutely right that >5% of records relating to Black African babies did not link to HES. The text has been revised to reflect this.

Discussion

Comment: In the first paragraph the authors state that 4-6% of birth registration/notification records were 'incorrectly linked' to HES delivery records. Can this statement be clarified?

Response: This has now been clarified in the paper.

Comment: A reference 14 is cited but this is missing from the reference list.

Response: Reference 14 from text has now been removed.

Conclusions

Comment: The second paragraph may well be true but it is not supported by results presented in this paper. Please either provide supporting results or amend.

Response: This paragraph has now been removed.

Tables

Comment: Should the heading of the 3rd column in Table 1 read including (rather than excluding) duplicated HES delivery records?

Response: Number of duplicate HES records are shown in the 4th column so the heading for the 3rd column is correct i.e. it excludes duplicate HES delivery records.

Comment: In general the headings of the tables could be tidied up and made more consistent. For example, Table 3 does not provide results by match rank and the terminology for specific datasets is variable eg birth notification/NN4B.

Response: Thanks for noting this. I have now made the headings in all the tables consistent.

Comment: No reporting checklist accompanies this paper as far as I can tell. A STROBE or, preferably, a RECORD statement would be appropriate.

Response: This is part of a larger study, so the article does not contain all the elements mentioned in the RECORD statement. A RECORD statement completed as far as possible is attached

Response to comments from Reviewer: 2

Reviewer Name: Toan C Ong

Comment: The value of linking real patient data is apparent. However, that alone will be insufficient to justify a separate publication. I struggled to see the scientific contributions of this paper. This entire paper can fit into the Result and Discussion of a more comprehensive paper.

Response: The datasets concerned are key national datasets and this is the first time they have been linked in this way. The linkage is part of a larger study with a number of analyses, so there will be more than one article describing these.

Comment: Other than the description of the linkage variables and how they were compared (exact or partial), the record linkage method was not adequately described.

Response: The linkage algorithm is described in the Appendix and Table 3 shows number of records linked at each step of the algorithm which is discussed in the Results section.

Comment: It was not clear how the linkage result was validated. The authors mentioned that process is described elsewhere but I disagree with that separation.

Response: The quality assurance process is very detailed and lengthy. For that reason the quality assurance has been described in a separate article by Gill Harper titled 'Quality assuring linked ONS birth records and maternity Hospital Episode Statistics delivery records on singleton and multiple births in England 2005 to 2014' but submitted to the same journal, in the hope that the two articles can be published together.

Comment: Many observations were described vaguely without further articulation. For example, quote from the paper "The linkage rate was 3 per cent lower for multiple births, 2 per cent lower in mother's aged under 15 and 3 per cent lower for those aged 40 years and above." Questions: Lower than what? Why lower?

Response: The linkage was 3 per cent lower for multiple births compared to singleton births, 2 per cent lower for mothers aged under 15 and 3 per cent lower for mothers aged 40+ compared to all other age groups. It is not known why this was the case but this findings are important when analysing these data.

Comment: Using the NHS number helped link the majority of the datasets. It is interesting to know how many records were linked using just the NHS number as the linkage variable. What is the performance when the NHS is absent?

Response: Step 8 of the algorithm used the NHS number only for linkage and we found there were issues with the quality of linkage. This suggests that there are issues with the quality of the recording of the NHS number on some administrative datasets. The number of records linked in steps 6 and 7 of the algorithm without the NHS number is lower compared to all other steps involving use of the NHS number, as shown in Table 2.

Comment: Overall, I find that the findings of this paper are very interesting. However, the authors should consider a comprehensive paper which discusses in details the linkage methods, the results, and the validation process. Such paper will be much more impactful than separate ones.

Response: We hope that this linkage paper will be published at the same time as the quality assurance paper which was submitted to the Journal at the same time.

VERSION 2 – REVIEW

REVIEWER	Rachael Wood NHS National Services Scotland, Information Services Division, Scotland, UK I am responsible for maintaining the linkage of maternal and baby records in Scotland but have no involvement in this English work
REVIEW RETURNED	31-Aug-2017

GENERAL COMMENTS	I am broadly happy that my previous comments have been adequately addressed. I continue to think that this is important work that should be published. The only residual comment I would make is that I still find the issue of linkage to duplicate HES records somewhat confusing. I presume this refers to the situation when the same ONS birth record links to multiple HES delivery (or birth) records. It is not clear how these were dealt with eg how the correct link to retain was identified (or were all these birth records considered unlinked?). No specific results on linkage to multiple HES birth records are presented in Table 3 but then there is a comment in the penultimate paragraph of the Discussion suggesting that linkage to multiple HES birth records was a significant problem. Finally, in Appendix A it would be helpful to clarify what Eptype 2, 5, 3, and 6 refer to for users not familiar with HES data.
--

REVIEWER	Toan C Ong University of Colorado Anschutz Medical Campus
REVIEW RETURNED	05-Sep-2017

GENERAL COMMENTS	Thanks for revision. I have no additional comments.
---

VERSION 2 – AUTHOR RESPONSE

Many thanks for the reviewers comments. I have addressed further comments from Rachael Wood in the revised draft as follows:

Point 1: The only residual comment I would make is that I still find the issue of linkage to duplicate HES records somewhat confusing. I presume this refers to the situation when the same ONS birth record links to multiple HES delivery (or birth) records. It is not clear how these were dealt with eg how the correct link to retain was identified (or were all these birth records considered unlinked?)

Response: Sorry for not being clear about this in my draft. This is clarified in Gill Harper's paper which we are hoping will be published at the same time. But I have now added a paragraph in my paper stating that where the same ONS birth record was linked to multiple HES delivery records, during assessment of the quality of linkage, the HES delivery record with mother's and baby's information matching the ONS birth record and with greatest amount of information on onset of labour and method of delivery was retained for analysis. The other records were discarded.

Point 2: No specific results on linkage to multiple HES birth records are presented in Table 3 but then there is a comment in the penultimate paragraph of the Discussion suggesting that linkage to multiple HES birth records was a significant problem.

Response: Although we did not assess the quality of linkage, our previous linkage study showed that there were many duplicate HES birth records linked to ONS birth records. Also a high proportion of baby records are known to be missing in Maternity HES.

Point 3: in Appendix A it would be helpful to clarify what Epitype 2, 5, 3, and 6 refer to for users not familiar with HES data.

Response: I have now added definitions of Epitype 2,3,5 and 6 to the Appendix.